# The Image Definition Assessment of Optoelectronic Tracking Equipment Based on the BRISQUE Algorithm with Gaussian Weights

**DOI:** 10.3390/s23031621

**Published:** 2023-01-30

**Authors:** Ning Zhang, Cui Lin

**Affiliations:** 1Changchun Institute of Optics, Fine Mechanics and Physics, Chinese Academy of Sciences, Changchun 130033, China; ning0025@163.com; 2University of Chinese Academy of Sciences, Beijing 100049, China

**Keywords:** optoelectronic tracking equipment, image definition, defocus, BRISQUE algorithm, support vector machine

## Abstract

Defocus is an important factor that causes image quality degradation of optoelectronic tracking equipment in the shooting range. In this paper, an improved blind/referenceless image spatial quality evaluator (BRISQUE) algorithm is formulated by using the image characteristic extraction technology to obtain a characteristic vector (CV). The CV consists of 36 characteristic values that can effectively reflect the defocusing condition of the corresponding image. The image is evaluated and scored subjectively by the human eyes. The subjective evaluation scores and CVs constitute a set of training data samples for the defocusing evaluation model. An image database that contains sufficiently many training samples is constructed. The training model is trained to obtain the support vector machine (SVM) model by using the regression function of the SVM. In the experiments, the BRISQUE algorithm is used to obtain the image feature vector. The method of establishing the image definition evaluation model via SVM is feasible and yields higher subjective and objective consistency.

## 1. Introduction

The image, which is an important carrier of information, has been widely used in health, medical community, consumer electronics, etc. However, distortions are inevitably induced during image acquisition, transmission, processing, and display. The distortions cause the image quality degradation [1]. Evaluating, comparing, and optimizing the image quality effectively has gradually become a research hotspot in many fields, such as visual psychology, image processing, pattern recognition, and artificial intelligence [2,3,4].

Image distortion occurs, to a certain extent, in the process of acquisition, processing, compression, transmission, and display. Therefore, it is necessary to establish objective and effective quality assessment methods to evaluate the image quality [5,6,7]. At present, the image quality assessment includes subjective assessment and objective assessment. Image quality is evaluated by the subjective perception of the human eyes in a subjective evaluation method. As an objective evaluation method of the image quality, the mathematical models of image quality assessment are established [8,9].

The objective methods of image quality assessment include full reference image quality assessment (FR-IQA), reduced reference image quality assessment (RR-IQA), and no reference image quality assessment (NR-IQA), according to whether the reference image is needed. In the paper, NR-IQA is used to evaluate the image quality [10,11].

The main factors which affect the quality of optical measurement images include atmospheric disturbance, atmospheric extinction, optical diffraction of optical lens, defocusing, image motion, camera jitters, noise of image sensor, and so on. The image quality assessment method of defocused image is mainly studied in this paper.

If the external noises can be ignored, defocus is an important factor in image blur in the image tracking process of optoelectronic tracking equipment. To estimate the defocus severity of optoelectronic tracking equipment, image quality is evaluated objectively via an image quality evaluation algorithm [12,13]. At the same time, image characteristic values, which reflect the image quality, are obtained. The values can provide a condition for establishing the model for evaluating the focus performance based on the correlations between image characteristic values and defocus state parameters.

The optical system of optoelectronic tracking equipment can be regarded as a low-pass filter and an increase in the defocus is equivalent to a reduction in the filter cut-off frequency [14,15,16,17,18].

This paper mainly studies image evaluation indices in the defocusing state of optoelectronic tracking equipment and a method for obtaining the image characteristic values based on the indices. The characteristic values that are obtained via an image evaluation algorithm can be used to repair the image quality degradation that is caused by defocused equipment. The result of the image evaluation algorithm should be consistent with the subjective perception of the human eyes [19,20,21].

The causes of image blur also include interference factors, such as image motion of equipment and data compression, in addition to the defocus of the imaging system. A general referenceless image evaluation algorithm should be selected instead of a referenceless image evaluation algorithm with known distortion [22,23,24].

Comparisons are performed from two aspects: the theory and the performance of the evaluation algorithm. The main referenceless image quality evaluation algorithms that perform well are as follows: (1) Moorthy’s blind image quality index (BIQI) algorithm, which is implemented in the wavelet domain [25]; (2) Moorthy’s distortion-identification-based image verity and integrity evaluation (DIIVINE) algorithm, which is based on the BIQI algorithm [7]; (3) Saad’s distortion-identification-based image verity and integrity evaluation (DIIVINE) algorithm [26] and the BLIINDS-II improved algorithm [27]; (4) Mittal’s BRISQE algorithm [28] and the natural image quality evaluator (NIQE) algorithm, which is referenceless [29]; (5) Li’s general regression neural network (GRNN) algorithm [30]; and (6) Lintao Han’s combining convolution and self-attention for image quality assessment network [31].

Spatial distortion directly affects the visual quality of an image. By considering effective spatial characteristics, image quality evaluation can achieve increased consistency with subjective evaluation. At the same time, the characteristic values that are obtained via spatial characteristic extraction lay the foundation for the study of building an evaluation model for the defocused state. 

Ruderman et al. found that the luminance of natural image normalization tends to follow a normal (Gaussian) distribution [32]. They posit that the distortion of an image changes the statistical characteristics of the normalization coefficient. By measuring the changes in the statistical characteristics, the distortion type can be predicted and the image visual quality can be evaluated [33]. Based on this theory, Mr. Mittal put forward the BRISQUE algorithm [28], which is based on the image spatial statistical characteristics. Ronin Institute et al. apply a broad spectrum of statistics of local and global features to characterize the variety of possible video distortions [34].

Based on the image defocus characteristics of optoelectronic tracking equipment in this paper, an improved BRISQUE algorithm is used with image characteristic extraction technology to obtain a characteristic value (CV). The CV includes 36 characteristic values that effectively reflect the defocus condition of the image [35]. The image is evaluated by the human eyes and scored subjectively. Subjective evaluation scores and feature vectors constitute a set of training data samples of the defocus evaluation model. A sufficient amount of training samples is obtained by calculating the CVs of the image database. Then, the evaluation model is obtained by using a machine learning method that is based on SVM to train the samples [36]. 

Many studies have employed machine learning models for prediction or classification in many fields. A convolutional neural network (CNN) is used for robust classification of PV panel faults [37]. A support vector machine (SVM) has become a common method of discrimination. In the field of machine learning, it is usually used for pattern recognition, classification, and regression analysis. For example, CNN- and SVM-based models can provide doctors with the detection of heart failure using electrocardiogram signals [38]. The SVM and general regression neural networks (GRNN) were used for the diagnosis of malfunction [39]. The adaptive support vector machine (A-SVM) was introduced for classification together with the ORICA-CSP method [40].

The defocused image sequences of the optoelectronic equipment are computed via the BRISQUE algorithm to obtain the CVs. The CVs are inputted into the evaluation model to calculate the prediction scores. The image sequences are evaluated by the human eyes subjectively. By considering the subjective and objective consistency of the results of the evaluation algorithm, the effectiveness of the evaluation algorithm is assessed.

## 2. Acquiring the CV via the Improved BRISQUE Algorithm

The image database is built and the CVs of image samples from the image database are obtained via the improved BRISQUE algorithm, which is weighted by a Gaussian function. The image samples are evaluated subjectively by the human eyes and used as SVM model training samples.

### 2.1. Training Image Sample Selection and Database Establishment

Many preliminary studies and experiments have demonstrated that if an image sequence of the optoelectronic tracking equipment is used for training directly, the training model will be inaccurate, which will lead to the failure of forecast evaluation. The main reason is that it is impossible to cover various details because the target and background tracking are too monotonous. Using public database images for training is proposed. We have used three public databases, namely, Laboratory for Image & Video Engineering (IVE), Categorical Subjective Image Quality (CSIQ), and Tampere Image Database (TID2013). Table 1 lists the databases that are used in this article and their data types.

According to the defocus characteristics of the device tracking image, an image database that includes images in a sequence that ranges from defocused to focused and back to defocused is established and each image is subjectively evaluated and scored. The scoring principle is that a severely defocused image is assigned a low score and a better focused image has a higher score. The results of model training demonstrate that the size of the database should exceed 1000 pictures and the quality of the database directly affects the application stability.

### 2.2. BRISQUE Algorithm

Two important advantages of using the BRISQUE algorithm are that the image definition evaluation score that is obtained by the algorithm can effectively reflect the defocus state, and the obtained image characteristic vector facilitates the subsequent training and evaluation of the machine learning model.

From an image, the BRISQUE algorithm is used to extract 36 characteristic values, which include the variances of the image brightness and the mean value. These features are called local normalized brightness statistical characteristics.

Given an intensity image *I*(*i*,*j*), an operation that subtracts the image mean can be applied to the image to obtain the mean subtracted contrast normalized (MSCN) image *Î*(*i*,*j*):
(1)
$$\hat{I}\left(i,j\right)=\frac{I\left(i,j\right)-\mu \left(i,j\right)}{\sigma \left(i,j\right)+C}$$

where *i* = 1, …, *M* and *j* = 1, …, *N* are spatial indices; *M* and *N* are the image height and width, respectively; *C* is a constant that prevents instabilities from occurring when the denominator tends to zero; and *μ*(*i*,*j*) and *σ*(*i*,*j*) are the local mean and standard deviation, respectively, of *I*(*i*,*j*).

We model the statistical relationship between neighboring pixels using the empirical distributions of the pairwise products of neighboring MSCN coefficients along four orientations: horizontal (*H*), vertical (*V*), main diagonal (*D*1), and secondary diagonal (*D*2).

(2)
$$H\left(i,j\right)=\hat{I}\left(i,j\right)\hat{I}\left(i+1,j\right)$$


(3)
$$V\left(i,j\right)=\hat{I}\left(i,j\right)\hat{I}\left(i,j+1\right)$$


(4)
$$D1\left(i,j\right)=\hat{I}\left(i,j\right)\hat{I}\left(i+1,j+1\right)$$


(5)
$$D2\left(i,j\right)=\hat{I}\left(i,j\right)\hat{I}\left(i+1,j-1\right)$$


The statistical properties of the MSCN coefficients are affected by the presence of distortion. Quantifying these changes will make it possible to predict the type of distortion that affects an image and its perceptual quality. According to [24], a generalized Gaussian distribution (GGD) can be used to effectively capture a broader spectrum of distorted image statistics. The GGD with zero means is expressed as follows:
(6)
$$f\left(x;\alpha ,\sigma^{2}\right)=\frac{\alpha }{2\beta \Gamma \left(1/\alpha \right)}\exp\left(-{\left(\frac{\left|x\right|}{\beta }\right)}^{\alpha }\right)$$

where

(7)
$$\beta =\sigma \sqrt{\frac{\Gamma \left(1/\alpha \right)}{\Gamma \left(3/\alpha \right)}}$$

and Γ(*·*) is the gamma function:
(8)
$$\begin{array}{cc} \Gamma \left(a\right)=\int_{0}^{\infty }t^{a-1}e^{-t}dt & a>0 \end{array}$$


The shape parameter, which is denoted as *α*, controls the ‘shape’ of the distribution, while *σ*^2^ control the variance. The parameters of the GGD (*α*,*σ*^2^) are estimated via the moment-matching-based approach that was proposed in [41].

The appropriate values of α and σ are calculated via the moment-matching-based method and are two of the 36 characteristic values to be obtained. The parameters (*ν*,*σ_l_*,*σ_r_*) and *η* are calculated based on Equations (9) and (12) for the other four images: *H*, *V*, *D*1, and *D*2.

(9)
$$f\left(x;\alpha ,\sigma^{2}\right)=\{\begin{array}{c} \begin{array}{cc} \frac{\nu }{\left(\beta_{l}+\beta_{r}\right)\Gamma \left(1/\nu \right)}\exp\left(-{\left(\frac{-x}{\beta_{l}}\right)}^{\nu }\right), & x<0 \end{array}\\ \begin{array}{cc} \frac{\nu }{\left(\beta_{l}+\beta_{r}\right)\Gamma \left(1/\nu \right)}\exp\left(-{\left(\frac{-x}{\beta_{r}}\right)}^{{}^{\nu }}\right), & x\geq 0 \end{array} \end{array}$$

where

(10)
$$\beta_{l}=\sigma_{l}\sqrt{\frac{\Gamma \left(1/\nu \right)}{\Gamma \left(3/\nu \right)}}$$


(11)
$$\beta_{r}=\sigma_{r}\sqrt{\frac{\Gamma \left(1/\nu \right)}{\Gamma \left(3/\nu \right)}}$$


(12)
$$\eta =\left(\beta_{r}-\beta_{l}\right)\sqrt{\frac{\Gamma \left(2/\nu \right)}{\Gamma \left(1/\nu \right)}}$$


The details of the calculation process are presented in [24]. Via Equations (2)–(12), we obtain 16 + 2 = 18 characteristic values. The other 18 characteristic values must be calculated in other ways. The original image is down-sampled with a sampling

The characteristic values of the down-sampled image are calculated by following the given steps again and we obtain another 18 characteristic values. Now, the calculation of the 36 characteristic values is complete.

### 2.3. Improved BRISQUE Algorithm That Is Weighted by a Gaussian Function

Preliminary model training and prediction studies demonstrate that the characteristic values that were directly obtained via the BRISQUE algorithm cannot stably evaluate the defocused image sequence. For this particular situation, an improved BRISQUE algorithm that is weighted by a Gaussian function is selected in this paper.

The pixels of the training image are scanned by using a Gaussian function template and the center pixel value of the template is replaced with the weighted average gray value of the pixels in the neighborhood that is determined by the template. The template parameters of the Gaussian function are shown in Table 2. The image that is obtained by weighting the training image by the Gaussian function is denoted as VarI. The characteristic values of the new image are calculated by following the specified steps and we obtain 36 characteristic values, which are the input of machine learning training.

## 3. Support Vector Machine Model and Training

SVM is one of the basic methods of machine learning and the most important branch of machine learning theory [42,43,44]. It plays an important role in the practical applications of machine learning. SVM, which is a supervised learning model, is commonly used for pattern recognition, classification, and regression analysis.

This paper uses the regression function of SVM. The improved BRISQUE algorithm is used to calculate the CVs and subjective evaluation scores of images in the image database as the model training sample for obtaining the SVM model. The image definition CVs of the image database, which are calculated via the improved BRISQUE algorithm, are the independent variables. The scores of the subjective evaluation are the dependent variables. The independent and dependent variables are used as model training samples to obtain the SVM model. The image CVs of optoelectronic tracking equipment are input into the SVM model and predicted to obtain image evaluation scores. By comparing with the subjective evaluation of the human eyes, the accuracy and reliability of the evaluation are assessed. If the evaluation result does not meet the requirements, the above process can be iterated until a subjective and objective SVM evaluation model is obtained. Another image database can be used to calculate the characteristic vectors as needed and the image quality is scored for the inputs of the new training model via SVM.

This paper calls the LIBSVM library function, which was developed by Professor Chi-Jen Lin [45] to train and test the SVM model. The LIBSVM library function version is libsvm-3.23. In this paper, the support vector regression model “ε-SVR” is used in SVM.

The specified training sample can be represented as {(***x***_1_,*z*_1_),……,(***x****_l_*,*z_l_*)}, where ***x****_i_* ∈ *R^n^* is the characteristic vector, which is obtained via the improved BRISQUE algorithm and composed of 36 characteristic values, and *z_i_* ∈ *R*^1^ denotes the subjective evaluation score of the image, which is the target output of the training model. When the penalty parameter *C* > 0 and the parameter *ε* > 0, the standard form of the SVR is as expressed in Equation (13):
(13)
$$\underset{w,b,\xi ,\xi^{*}}{\min}\left(\frac{1}{2}w^{T}w+C\sum_{i=1}^{l}\xi_{i}+C\sum_{i=1}^{l}\xi_{i}^{*}\right)$$

s.t.

(14)
$$w^{T}\phi \left(x_{i}\right)+b-z_{i}\leq \varepsilon +\xi_{i}$$


(15)
$$z_{i}-w^{T}\phi \left(x_{i}\right)-b\leq \varepsilon +\xi_{i}$$


(16)
$$\xi_{i},\xi_{i}^{*}\geq 0,i=1,\cdots \cdots ,l$$


According to the principle of SVM, Equation (13) is converted to a dual problem to calculate α. The radial basis function (RBF) is selected as the kernel function, which is denoted as ***Κ***(***x***,***z***) = *ϕ^T^*(***x***)*ϕ*(***z***); the form of the RBF is as follows:
(17)
$$Κ\left(\left\|x-z\right\|\right)=e^{-\frac{{\left\|x-z\right\|}^{2}}{{\left(2\times \sigma \right)}^{2}}}$$

where *σ* is set to 0.5.

The training parameters of the LIBSVM library function are set as follows: penalty parameter *C* is set to 1024, the probability estimate is set to 1, and other parameters use the default parameter values of the LIBSVM function.

The samples from the image database of Table 1 are input into the SVM model and model training is completed. The number of support vectors, which is denoted as *total_sv*, is 772, and the bias *b* is −118.247.

## 4. Defocused Image Acquisition and Image Evaluation Test

### 4.1. Defocused Image Sequence Acquisition

In the process of tracking the real target using the optoelectronic tracking equipment, to ensure that the target can be tracked effectively, the focus state cannot be adjusted. The acquired image samples typically do not contain all image definition features, which makes it impossible to fully evaluate the performance of the SVM model.

To identify the test images that meet the requirements, in the process of evaluating the imaging quality of the optoelectronic tracking device, an imaging system is built for obtaining image samples of various defocus states. A photo of the system is shown in Figure 1. A Nikon 800 mm/F5.6 fixed-focus lens from the Nikon Corporation of Japan is used in the imaging system. The piA2400-17 visible light camera is from BASLER Corporation of Germany. The main properties of the camera are as follows: pixel size: 3.45 μm × 3.45 μm; and the number of pixels: 2448 × 2050.

### 4.2. Predictive Test of Definition Evaluation of Defocused Images

In this paper, a series of defocused and focused images with continuous change were obtained by manually controlling the defocused position of the optical lens in the imaging system. The images are used to test the effectiveness of the definition evaluation algorithm of defocused images. At the same time, they are also used for algorithm comparison.

To acquire stable evaluation scores, static scenes are photographed using the imaging system. Therefore, the image sequences in this paper are very similar to human visual perception. The major differences between the images are definition and edge sharpness. Serial numbers of the clear images are given in advance.

The image sequences are inputted into the trained SVM model, and the image definition evaluation scores of defocused image sequences are the outputs of SVM model. Because the image-focusing process and the serial numbers of the clear images are known, the image definition evaluation scores can be compared with the defocused states of the image sequences.

For the image sequences, the larger the score, the clearer the image is. Due to the evaluation scores related to the CVs obtained by the BRISQUE algorithm, they are not fixed values. However, the scores can reflect the definition of the image sequence with the same scene. The image definition scores vary greatly among the image sequences with different scenes.

#### 4.2.1. Single-Peak Defocused Image Test

The indoor image sequence that was obtained by the experimental imaging system is shown in Figure 2. The shooting process is from defocus to focus and back to defocus. The 9th image of the 12 images in Figure 2 has the best visual effect. In the predictive evaluation test of the 12 pictures via the SVM model, we obtained the curve that is shown in Figure 3. The *X*-axis of the curve represents the serial numbers of the pictures and the *Y*-axis represents the corresponding image definition evaluation values. The first image has the largest defocused position, and its evaluation score is only −3.34. The ninth image with the highest definition has the highest score of 20. The curve is consistent with the clarity of the real image.

#### 4.2.2. The Test of Algorithm Comparison

The structural similarity (SSIM) is compared with the SVM model in this paper. As shown in Figure 4, the first image in the image sequence has the largest defocus, and it is the most blurred image to human visual perception. As the serial number increases, the image has a higher definition with defocused decreasing. The 14th image is the clearest to human perception. The evaluation curves with SSIM and the SVM trained model are shown in Figure 5 and Figure 6, respectively. Due to the different calculation principles of the two algorithms, the evaluation scores cannot be directly compared.

As shown in Figure 5, the evaluation scores with SSIM increase monotonously in the range of the first image to the eleventh image, which is consistent with the subjective evaluation by human eyes. However, the evaluation scores start to fall from the 12th image, and it is inconsistent with subjective evaluation. As shown in Figure 6, the evaluation scores with the SVM model increase with the serial numbers of the images in the sequence. Image 1 has the lowest score of 10.4, and image 14 has the highest score of 65.5. The evaluation with SVM is completely consistent with human subjective evaluation.

#### 4.2.3. Dual-Peak Defocused Image Test

The dual-peak defocused image sequence is shown in Figure 7. The shooting process is focus, defocus, focus, and defocus in image 8 and image 21, respectively. According to the predictive evaluation test of the 28 pictures using the SVM model, the curve in Figure 8 is obtained. The *X*-axis of the curve represents the serial numbers of the pictures and the *Y*-axis represents the corresponding image definition evaluation values. We marked the two focused peak images with red hexagonal stars. The score of the 8th image is 58.3, and the score of the 22nd image is 55. The curve is consistent with the subjective evaluation by the human eyes of the test images. The curve also exhibits dual peaks, which demonstrates the convergence of the prediction model.

#### 4.2.4. Repeatability Testing of Dual-Peak Defocused Image

Repeated tests were carried out to check the generalization performance of the SVM model. Another 29 images were acquired by changing the imaging scene and imaging process. The images were captured in order of focus, defocus, focus, defocus, and focus. Two randomly selected images in this image sequence are shown in Figure 9, and the definition evaluation scores of the sequence with the SVM model are shown in Figure 10. The result shows that the evaluation scores change by the focusing and defocused order, and the definition evaluation with the SVM model shows stability consistent with perspective evaluation. The SVM model has good generalization performance.

Through many test experiments, the image feature characteristic vectors are calculated via the improved BRISQUE algorithm and the evaluation model that is established via the SVM algorithm is used to evaluate the definition evaluation prediction. The evaluation results are highly consistent with the subjective evaluation results of the human eyes.

## 5. Discussion

For the purpose of increasing the effect of model training, we improved the BRISQUE algorithm that is weighted by the Gaussian function, and other weighted functions and parameters can also be researched in the future. The kernel function is selected as the radial basis function (RBF) in this paper, and other kernel functions can also be tried. In the future, the research of image objective evaluation model training based on machine learning will focus on two aspects. First, we should research and improve the new methods, which is to characterize the image spatial statistical characteristics. Second, we can introduce new machine learning algorithms, such as deep learning algorithms, which lead to a model with stronger self-learning ability.

## 6. Conclusions

Aiming at the problem of defocusing on large-scale optoelectronic tracking equipment in the shooting range, the use of image definition indicators for evaluation is proposed in this paper. An improved BRISQUE algorithm is used to objectively evaluate a defocused image and a CV that consists of 36 characteristic values are obtained. The CV is input into a previously trained SVM model to obtain an image definition evaluation score. Many image samples were obtained using the established imaging experimental system and experimental tests were carried out. The experimental results demonstrate that the image definition evaluation method that is used in this paper can effectively evaluate the defocusing condition of an optoelectronic tracking device, and the obtained image CV can effectively reflect the image defocus state.

## Figures and Tables

**Figure 1 sensors-23-01621-f001:**
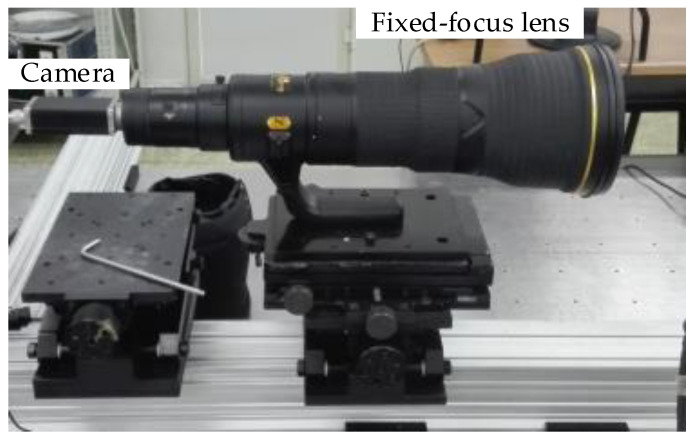
Photo of the imaging system.

**Figure 2 sensors-23-01621-f002:**
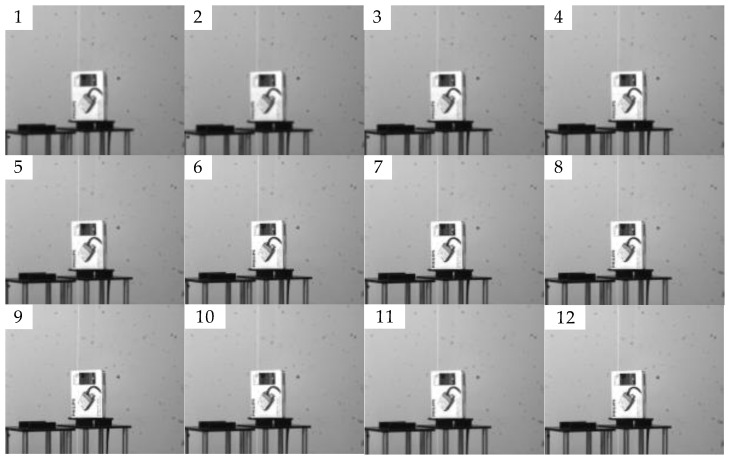
Indoor defocused image sequence. Sub-figures (1–12) represent the imaging results of the laboratory imaging system for the same target. The shooting process is from defocus to focus and back to defocus. The 9th picture shows the focus state.

**Figure 3 sensors-23-01621-f003:**
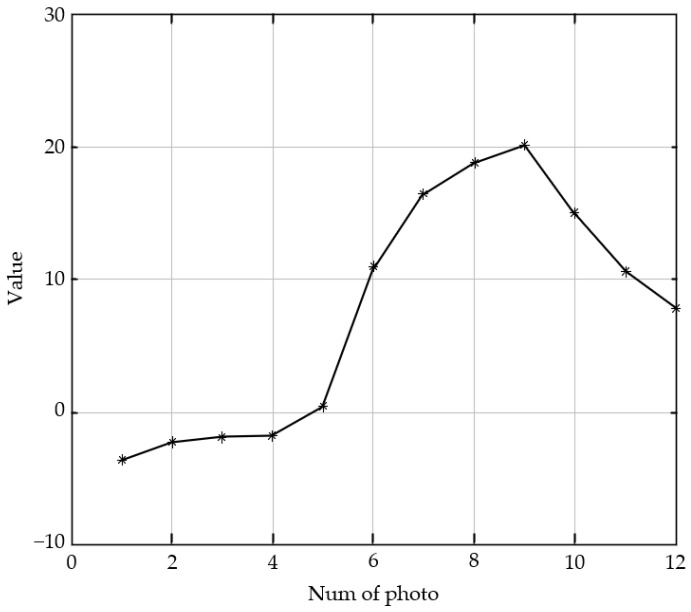
Predictive scores of the defocused image sequence.

**Figure 4 sensors-23-01621-f004:**
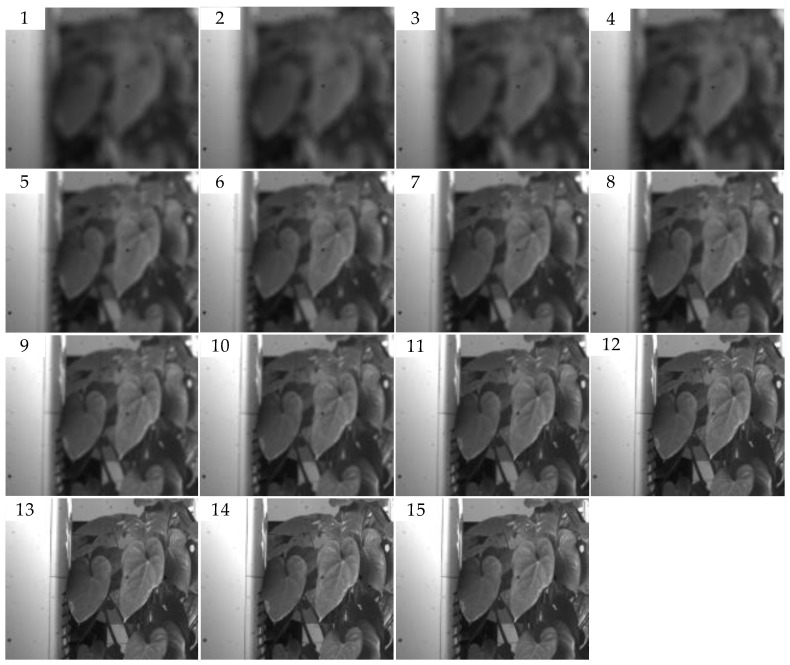
Defocused image sequence for comparison. Sub-figures (1–15) shows the imaging effect of the same target at different degrees of defocus. The first image is the most defocused, and the 15th image is the clearest.

**Figure 5 sensors-23-01621-f005:**
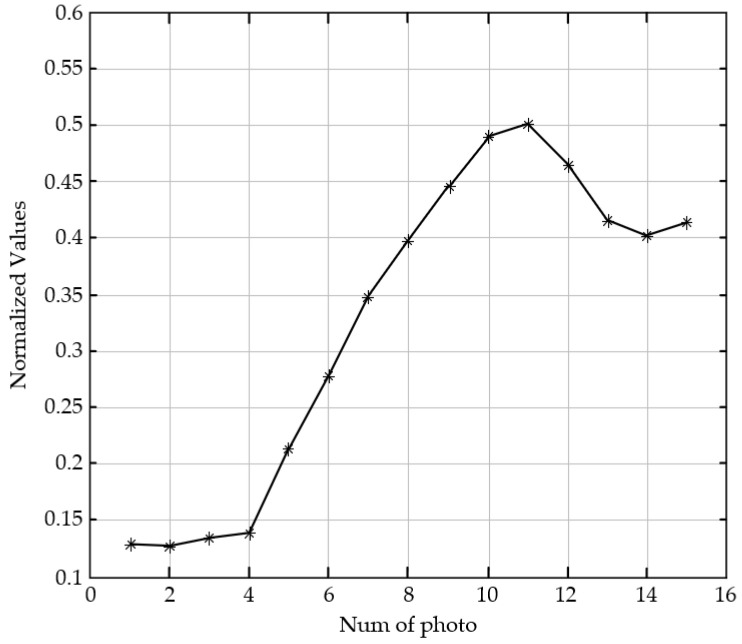
Evaluation scores of the image sequence in Figure 4 with SSIM.

**Figure 6 sensors-23-01621-f006:**
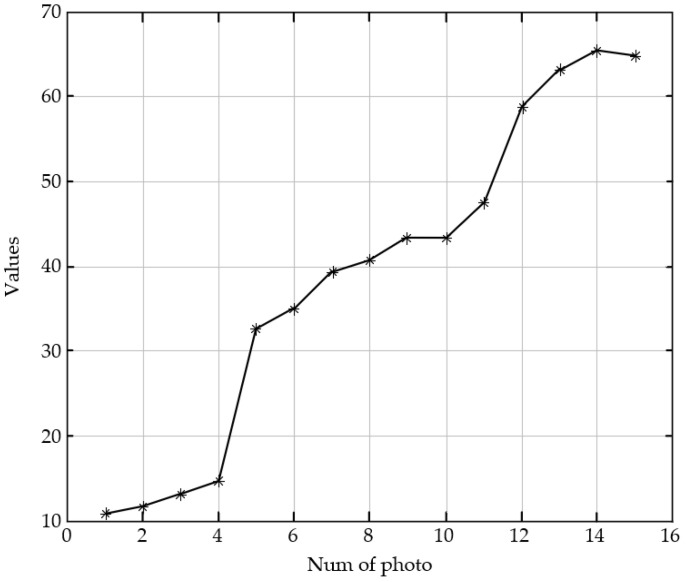
Evaluation scores of the image sequence in Figure 4 with the SVM model.

**Figure 7 sensors-23-01621-f007:**
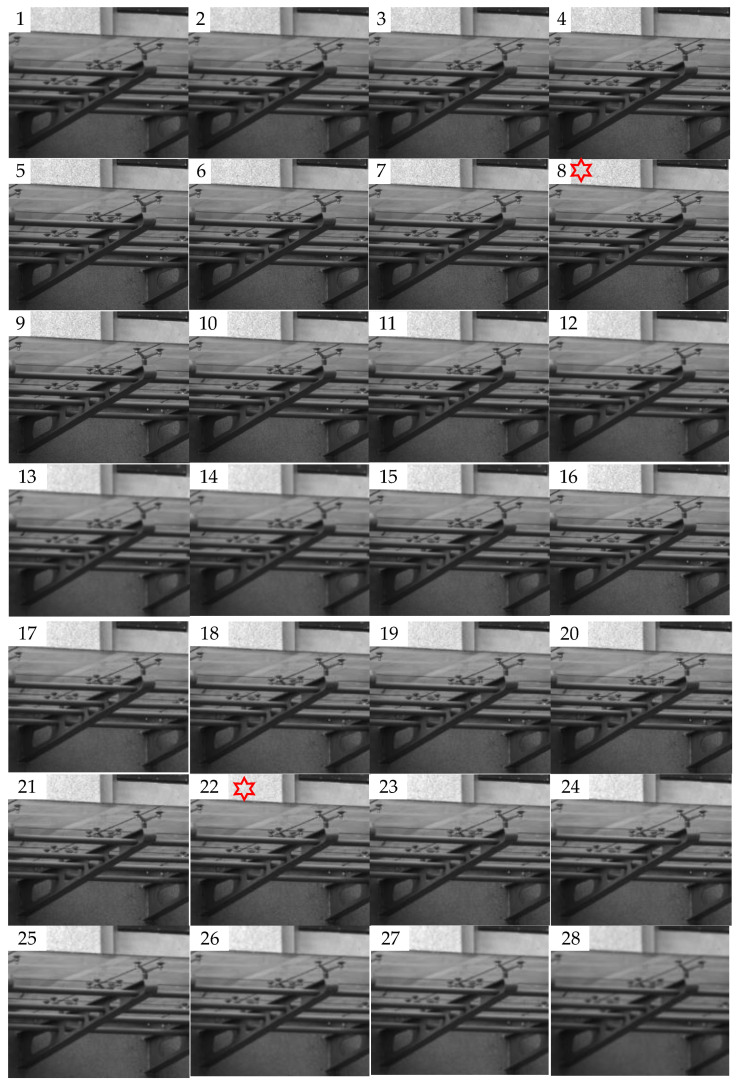
Dual-peak defocused image sequence (1–28). The shooting process is defocus, focus, defocus, focus, and defocus respectively. Image 8 and image 22 are in focus.

**Figure 8 sensors-23-01621-f008:**
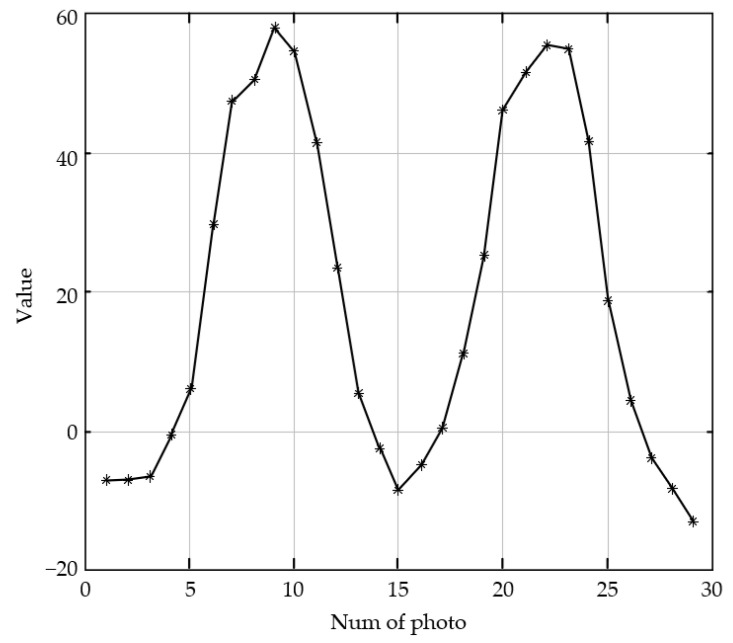
Predictive scores of the dual-peak defocused image sequence.

**Figure 9 sensors-23-01621-f009:**
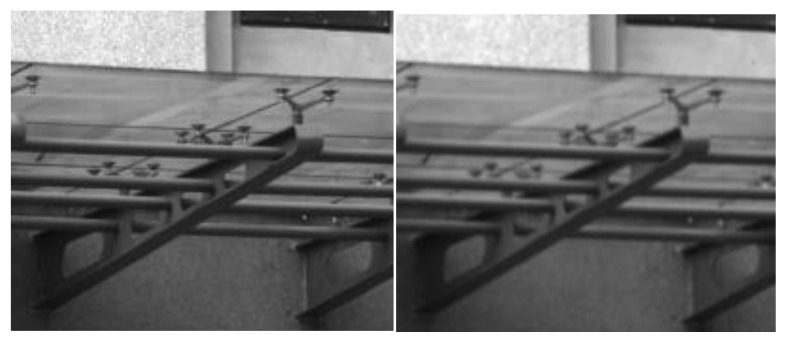
Defocused image sequence for repeatability testing (two randomly selected images).

**Figure 10 sensors-23-01621-f010:**
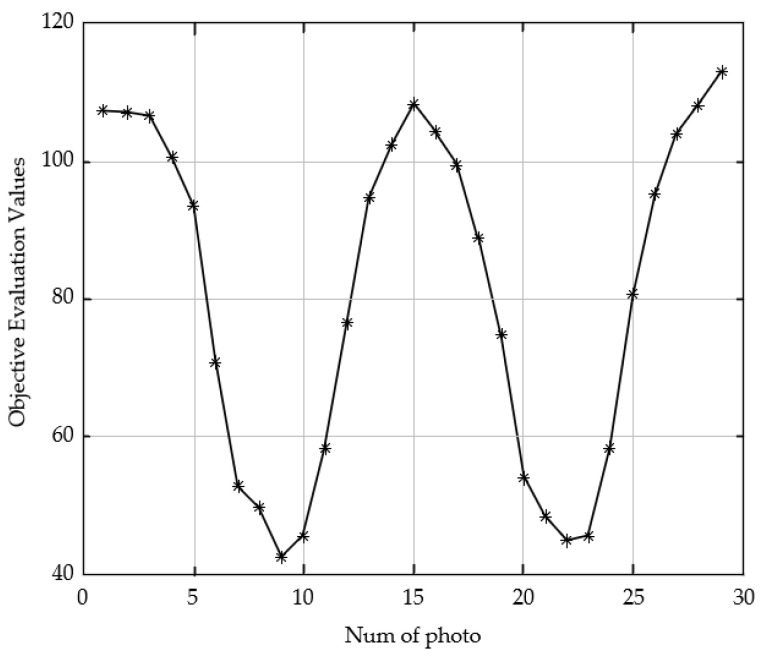
Evaluation scores with the SVM model for repeatability testing.

**Table 1 sensors-23-01621-t001:** Image databases for training the model.

Name	Num. of Distorted Images	Num. of Reference Images	Image Type
IVE	235	10	Grey and color images
TID2013	1700	25	Color images
CISQ	866	30	Color images

**Table 2 sensors-23-01621-t002:** Template of the weighted Gaussian function.

Weightiness	1	2	3	4	5	6	7
1	0.000157	0.00099	0.003	0.0043	0.003	0.00099	0.000157
2	0.00099	0.0062	0.0187	0.027	0.0187	0.0062	0.00099
3	0.0043	0.027	0.0813	0.1174	0.0813	0.027	0.003
4	0.003	0.0187	0.0563	0.0813	0.0563	0.0187	0.003
5	0.00099	0.0062	0.0187	0.027	0.0187	0.0062	0.00099
6	0.000157	0.00099	0.003	0.0043	0.003	0.00099	0.000157
7	0.000157	0.00099	0.003	0.0043	0.003	0.00099	0.000157

## Data Availability

Data are contained within the article.

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
