# Peer review of "The Image Definition Assessment of Optoelectronic Tracking Equipment Based on the BRISQUE Algorithm with Gaussian Weights"

_sensors, 2023, doi:10.3390/s23031621_

Round 1

Reviewer 1 Report

This paper presents a method to evaluate the image quality which can effectively reflect the defocusing condition. The proposed idea is reasonable and practical. The experiments using the pratical optoelectronic tracking equipment verify the consistency between the proposed evaluation method and the subjective image effect.

The detailed comments are as follows.
1. "human eye" should be "human eyes".
2. Can the numerator in Eq.(1) be negative?
3. In section 2.2, "...μ(i,j) and σ(i,j) are the local mean and variance" should be "...μ(i,j) and σ(i,j) are the local mean and standard deviation" .
4. In table 2, the "1" in the last tuple is "7"?
5. In section 3, "s.t. to" should be "subject to" or "s.t.".
6. It is suggested to give the comparative experimental results with existing methods.

Reviewer 2 Report

The authors expressed interesting research by implementing a theoretical approach and actual technology regarding optical image technology. However, the contents of the study are too insufficient to arouse interest outside researchers in the field.

This direction is contrary to the publication of papers that affect researchers in various fields pursued by Sensors journals. Therefore, in order for this thesis paper to be published, it is necessary to develop the paper in a more detailed and easy way to arouse the interest and understanding of various researchers. It is strongly recommended that the judges consider the following points.

1. The introduction is too specialized and the references are limited to the field. It is hoped that the results of the sensor reflecting the optical image principles and methods claimed by the authors must be added to the contents within the last 5 years, and the contents of the introductory section are greatly revised to make it accessible to researchers in various fields.

2. The main content of the results of this study seems to be Figures 2 and 4. Looking at successive images, it is difficult to understand the differences between the images unless you are a researcher in the field. It is necessary to briefly explain the meaning of each image.

3. Figures 3 and 5 show graphs for the interpretation of successive images. However, the explanation is too short. For example, what do the numbers on the y-axis mean? The maximum value of 3 is 20, whereas Figure 5 of the double peak is 60. Why do you see these differences? A scientific approach to this is too lacking. Please be sure to explain and provide references.

4. Finally, please explain the meaning of the change in the value of the y-axis in consecutive images. In the defocused image, it changes from the 5th and has the highest value in the 9th, but in the dual image, it changes from the 4th and has the 9th highest value, and it seems to change periodically. Is there any scientific basis for these changes in values? Please give a detailed explanation by taking a scientific approach to why this phenomenon appears. And, the reproducibility experiment is the most important in the paper. Please make sure to prove that the same tendency is shown even if you measure several times.

If modified to reflect the above, I believe that this paper will be able to become an excellent scientific paper on Sensors.
